# Computing Nash Equilibria in Generalized Interdependent Security Games

**Hau Chan**     **Luis E. Ortiz**
Department of Computer Science, Stony Brook University
{hauchan,leortiz}@cs.stonybrook.edu

## Abstract

We study the computational complexity of computing Nash equilibria in generalized interdependent-security (IDS) games. Like traditional IDS games, originally introduced by economists and risk-assessment experts Heal and Kunreuther about a decade ago, generalized IDS games model agents' voluntary investment decisions when facing potential direct risk and transfer-risk exposure from other agents. A distinct feature of generalized IDS games, however, is that full investment can reduce transfer risk. As a result, depending on the transfer-risk reduction level, generalized IDS games may exhibit strategic complementarity (SC) or strategic substitutability (SS). We consider three variants of generalized IDS games in which players exhibit only SC, only SS, and both SC+SS. We show that determining whether there is a pure-strategy Nash equilibrium (PSNE) in SC+SS-type games is NP-complete, while computing a single PSNE in SC-type games takes worst-case polynomial time. As for the problem of computing all mixed-strategy Nash equilibria (MSNE) efficiently, we produce a partial characterization. Whenever each agent in the game is indiscriminate in terms of the transfer-risk exposure to the other agents, a case that Kearns and Ortiz originally studied in the context of traditional IDS games in their NIPS 2003 paper, we can compute all MSNE that satisfy some ordering constraints in polynomial time in all three game variants. Yet, there is a computational barrier in the general (transfer) case: we show that the computational problem is as hard as the Pure-Nash-Extension problem, also originally introduced by Kearns and Ortiz, and that it is NP-complete for all three variants. Finally, we experimentally examine and discuss the practical impact that the additional protection from transfer risk allowed in generalized IDS games has on MSNE by solving several randomly-generated instances of SC+SS-type games with graph structures taken from several real-world datasets.

## 1 Introduction

*Interdependent Security (IDS) games* [1] model the interaction among multiple agents where each agent chooses whether to invest in some form of security to prevent a potential loss based on both direct and indirect (transfer) risks. In this context, an agent's *direct risk* is that which is not the result of the other agents' decisions, while *indirect (transfer) risk* is that which does.

Let us be more concrete and consider an application of IDS games. Imagine that you are an owner of an apartment. One day, there was a fire alarm in the apartment complex. Luckily, it was nothing major: nobody got hurt. As a result, you realize that your apartment can be easily burnt down because you do not have any fire extinguishing mechanism such as a sprinkler system. However, as you wonder about the cost and the effectiveness of the fire extinguishing mechanism, you notice that the fire extinguishing mechanism can only protect your apartment if a small fire originates in your apartment. If a fire originates in the floor below, or above, or even the apartment adjacent to yours, then you are out of luck: by the time the fire gets to your apartment, the fire would be fierce enough

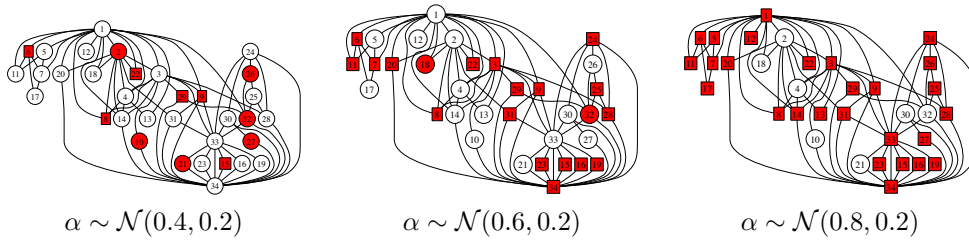

$\alpha \sim \mathcal{N}(0.4, 0.2)$      $\alpha \sim \mathcal{N}(0.6, 0.2)$      $\alpha \sim \mathcal{N}(0.8, 0.2)$

Figure 1: $\alpha$-IDS Game of Zachary Karate Club at a Nash Equilibrium. Legend: Square $\equiv$ SC player, Circle $\equiv$ SS player, Colored $\equiv$ Invest, and Non-Colored $\equiv$ No Invest

Table 1: Complexity of $\alpha$-IDS Games

| Game type | One PSNE | All MSNE | Pure-Nash Extension |
|:---:|:---:|:---:|:---:|
| SC <br> ($n$ SC players) | Always Exists <br> $O(n^2)$ | Uniform Transfers (UT) <br> $O(n^4)$ | NP-Complete |
| SS <br> ($n$ SS players) | Maybe Not Exist | UT wrt Ordering 1 <br> $O(n^4)$ | |
| SC + SS <br> ($n_{sc} + n_{ss} = n$) | NP-complete | UT wrt Ordering 1 <br> $O(n_{sc}^4 n_{ss}^3 + n_{sc}^3 n_{ss}^4)$ | |

already. You realize that if other apartment owners invest in the fire extinguishing mechanism, the likelihood of their fires reaching you decreases drastically. As a result, you debate whether or not to invest in the fire extinguishing mechanism given whether or not the other owners invest in the fire extinguishing mechanism. Indeed, making things more interesting, you are not the only one going through this decision process; assuming that everybody is concerned about their safety in the apartment complex, everybody in the apartment complex wants to decide on whether or not to invest in the fire extinguishing mechanism given the individual decision of other owners.

To be more specific, in the IDS games, the agents are the apartment owners, each apartment owner needs to make a decision as to whether or not to invest in the fire extinguishing mechanism based on cost, potential loss, as well as the direct and indirect (transfer) risks. The direct risk here is the chance that an agent will start a fire (e.g., forgetting to turn off gas burners or overloading electrical outlets). The transfer risk here is the chance that a fire from somebody else's (unprotected) apartment will spread to other apartments. Moreover, transfer risk comes from the direct neighbors and cannot be re-transferred. For example, if a fire from your neighbors is transferred to you, then, in this model, this fire cannot be re-transferred to your neighbors. Of course, IDS games can be used to model other practical real-world situations such as airline security [2], vaccination [3], and cargo shipment [4]. See Laszka et al. [5] for a survey on IDS games.

Note that in the apartment complex example, the fire extinguishing mechanism does not protect an agent from fires that originate from other apartments. *In this work, we consider a more general, and possibly also more realistic, framework of IDS games where investment can partially protect the indirect risk* (i.e., investment in the fire extinguishing mechanism can partially extinguish some fires that originate from others). To distinguish the naming scheme, we will call these *generalized IDS games* as $\alpha$-*IDS games* where $\alpha$ is a vector of probabilities, one for each agent, specifying the probability that the transfer risk will *not* be protected by the investment. In other words, agent $i$'s investment can reduce indirect risk by probability ($1$-$\alpha_i$). Given an $\alpha$, the players can be partitioned into two types: the SC type and the SS type. The *SC players* behave *strategic complementarily*: they invest if sufficiently many people invest. On the other hand, the *SS players* behave *strategic substitutability*: they do not invest if too many people invest.

As a preview of how the $\alpha$ can affect the number of SC and SS players and Nash equilibria, which is the solution concept used here (formally defined in the next section), Figure 1 presents the result of our simulation of an instance of SC+SS $\alpha$-IDS games using the Zachary Karate Club network [6]. The nodes are the players, and the edge between nodes $u$ and $v$ represents the potential transfers from $u$ to $v$ and $v$ to $u$. As we increase $\alpha$'s value, the number of SC players increases while the

number of SS players decreases. Interestingly, almost all of the SC players invest, and all of the SS players are "free riding" as they do not invest at the NE.

Our goal here is to understand the behavior of the players in $\alpha$-IDS games. Achieving this goal will depend on the type of players, as characterized by the $\alpha$, and our ability to efficiently compute NE, among other things. While Heal and Kunreuther [1] and Chan et al. [7] previously proposed similar models, we are unaware of any work on computing NE in $\alpha$-IDS games and analyzing agents' equilibrium behavior. The closest work to ours is Kearns and Ortiz [8], where they consider the standard/traditional IDS model in which one cannot protect against the indirect risk (i.e., $\alpha \equiv 1$). In particular, *we study the computational aspects of computing NE of $\alpha$-IDS games in cases of all game players being (1) SC, (2) SS, and (3) both SC and SS*. Our contributions, summarized in Table 1, follow.

- We show that determining whether there is a PSNE in (3) is NP-complete. However, there is a polynomial-time algorithm to compute a PSNE for (1). We identify some instances for (2) where PSNE does and does not exist.

- We study the instances of $\alpha$-IDS games where we can compute all NE. We show that if the transfer probabilities are uniform (independent of the destination), then there is a polynomial-time algorithm to compute all NE in case (1). Cases (2) and (3) may still take exponential time to compute all NE. However, based on some ordering constraints, we are able to efficiently compute all NE that satisfy the ordering constraints.

- We consider the general-transfer case and show that the pure-Nash-extension problem [8], which, roughly, is the problem of determining whether there is a PSNE consistent with some partial assignments of actions to some players, is NP-complete for cases (1), (2), and (3). This implies that computing all NE is likely as hard.

- We perform experiments on several randomly-generated instances of SC+SS $\alpha$-IDS games using various real-world graph structures to show $\alpha$'s effect on the number of SC and SS players and on the NE of the games .

## 2  $\alpha$-IDS games: preliminaries, model definition, and solution concepts

In this section, we borrow definitions and notations of *(graphical) IDS games* from Kearns et al. [9], Kearns and Ortiz [8], and Chan et al. [7]. In an $\alpha$-*IDS game*, we have an underlying *(directed) graph* $G = (V, E)$ where $V = \{1, 2, ..., n\}$ represents the $n$ *players* and $E = \{(i, j) | q_{ij} > 0\}$ such that $q_{ij}$ is the *transfer probability* that player $i$ will transfer the bad event to player $j$. As such, we define $\text{Pa}(i)$ and $\text{Ch}(i)$ as the set of parents and children of player $i$ in $G$, respectively.

In an $\alpha$-IDS game, each player $i$ has to make a decision as to whether or not to invest in protection. Therefore, the *action* or *pure-strategy* of player $i$ is binary, denoted here by $a_i$, with $a_i = 1$ if $i$ decides to invest and $a_i = 0$ otherwise. We denote the *joint-action* or *joint-pure-strategy* of all players by the vector $\mathbf{a} \equiv (a_1, \ldots, a_n)$. For convenience, we denote by $\mathbf{a}_{-i}$ all components of $\mathbf{a}$ except that for player $i$. Similarly, given $S \subset V$, we denote by $\mathbf{a}_S$ and $\mathbf{a}_{-S}$ all components of $\mathbf{a}$ corresponding to players in $S$ and $V - S$, respectively. We also use the notation $\mathbf{a} \equiv (a_i, \mathbf{a}_{-i}) \equiv (\mathbf{a}_S, \mathbf{a}_{-S})$ when clear from context.

In addition, in an $\alpha$-IDS game, there is a *cost of investment* $C_i$ and *loss* $L_i$ associated with the bad event occurring, either through direct or indirect (transfered) contamination. For convenience, we denote the *cost-to-loss ratio* of player $i$ by $R_i \equiv C_i / L_i$. We can parametrize the *direct risk* as $p_i$, the probability that player $i$ will experience the bad event from direct contamination.

Specific to $\alpha$-IDS games, the parameter $\alpha_i$ denotes the *probability of ineffectiveness of full investment in security (i.e., $a_i = 1$) against player $i$'s transfer risk*. Said differently, the parameter $\alpha_i$ models the degree to which investment in security can potentially reduce player $i$'s transfer risk. Player $i$'s *transfer-risk function* $r_i(\mathbf{a}_{\text{Pa}(i)}) \equiv 1 - s_i(\mathbf{a}_{\text{Pa}(i)})$, where $s_i(\mathbf{a}_{\text{Pa}(i)}) \equiv \prod_{j \in \text{Pa}(i)}[1 - (1 - a_j)q_{ji}]$, is a function of joint-actions of $\text{Pa}(i)$ because of the potential overall transfer probability (and thus risk) from $\text{Pa}(i)$ to $i$ given $\text{Pa}(i)$'s actions. One can think of the function $s_i$ as the *transfer-safety function* of player $i$. The expression of $s_i$ makes explicit the implicit assumption that the transfers of the bad event are independent. Putting the above together, the *cost function of player $i$* is

$$M_i(a_i, a_{Pa(i)}) \equiv a_i[C_i + \alpha_i r_i(a_{Pa(i)})L_i] + (1 - a_i)[p_i + (1 - p_i)r_i(a_{-i})]L_i \ .$$

Note that the safety function describes the situation where a player $j$ can only be "risky" to player $i$ if and only if $j$ does not invest in protection. We assume, without loss of generality (wlog), that $C_i \ll L_i$, or equivalently, that $R_i \ll 1$; otherwise, not investing would be a dominant strategy.

While a *syntactically* minor addition to the traditional IDS model, the parameter $\alpha$ introduces a major *semantic* difference and an additional complexity over the traditional model. The semantic difference is perhaps clearer from examining the best response of the players: player $i$ invests if

$$C_i + \alpha_i r_i(\mathbf{a}_{\mathrm{Pa}(i)})L_i < [p_i + (1-p_i)r_i(\mathbf{a}_{\mathrm{Pa}(i)})]L_i \Leftrightarrow R_i - p_i < (1 - p_i - \alpha_i)r_i(\mathbf{a}_{\mathrm{Pa}(i)}) \ .$$

The expression $(1 - p_i - \alpha_i)$ is positive when $\alpha_i < 1 - p_i$ and negative when $\alpha_i > 1 - p_i$. The best response condition flips when the expression is negative. (When $\alpha_i = 1 - p_i$, player $i$'s investment decision simplifies because the player's internal risk fully determines the optimal choice.)

In fact, the parameter $\alpha$ induces a partition of the set of players based on whether the corresponding $\alpha_i$ value is higher or lower than $1 - p_i$. We will call the set of players with $\alpha_i > 1 - p_i$ the set of *strategic complementarity (SC)* players. SC players exhibit as optimal behavior that their preference for investing *increases* as more players invest: they are "followers." The set of players with $\alpha_i < 1 - p_i$ is the set of *strategic substitutability (SS)* players. In this case, SS players' preference for investing *decreases* as more players invest: they are "free riders."

For all $i \in SC$, let $\Delta_i^{sc} \equiv 1 - \frac{R_i - p_i}{1 - p_i - \alpha_i}$; similarly for $\Delta_i^{ss}$, for $i \in SS$. We can define the *best-response correspondence* for player $i \in SC$ as

$$\mathcal{BR}_i^{sc}(\mathbf{a}_{\mathrm{Pa}(i)}) \equiv \begin{cases} 0, & \Delta_i^{sc} > s_i(\mathbf{a}_{\mathrm{Pa}(i)}), \\ 1, & \Delta_i^{sc} < s_i(\mathbf{a}_{\mathrm{Pa}(i)}), \\ [0,1], & \Delta_i^{sc} = s_i(\mathbf{a}_{\mathrm{Pa}(i)}) \ . \end{cases}$$

The *best-response correspondence $\mathcal{BR}_i^{ss}$ for player $i \in SS$* is similar, except that we replace $\Delta_i^{sc}$ by $\Delta_i^{ss}$ and "reverse" the strict inequalities above. We use the best-response correspondence to define NE (i.e., both PSNE and MSNE). We introduce randomized strategies: in a *joint-mixed-strategy* $\mathbf{x} \in [0,1]^n$, each component $x_i$ corresponds to player $i$'s probability of invest (i.e. $Pr(a_i = 1) = x_i$). Player $i$'s decision depends on *expected* cost, and, with abuse of notation, we denote it by $M_i(\mathbf{x})$.

**Definition** A joint-action $\mathbf{a} \in \{0,1\}^n$ is a *pure-strategy Nash equilibrium (PSNE)* of an IDS game if $a_i \in \mathcal{BR}_i(\mathbf{a}_{\mathrm{Pa}(i)})$ for each player $i$. Replacing $\mathbf{a}$ with a joint mixed-strategy $\mathbf{x} \in [0,1]^n$ in the equilibrium condition and the respective functions it depends on leads to the condition for $\mathbf{x}$ being a *mixed-strategy Nash equilibrium (MSNE)*. Note that the set of PSNE $\subset$ MSNE. Hence, we use NE and MSNE interchangably.

For general (and graphical) games, determining the existence of PSNE is NP-complete [10]. MSNE always exist [11], but computing a MSNE is PPAD-complete [12–14].

## 3   Computational results for $\alpha$-IDS games

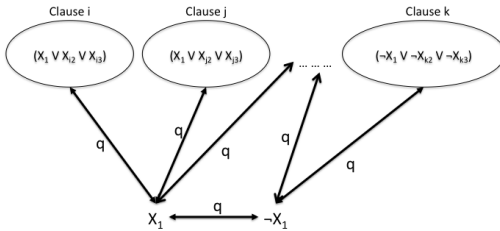

Figure 2: 3-SAT-induced $\alpha$-IDS game graph

In this section, we present and discuss the results of our computational study of $\alpha$-IDS games. We begin by considering the problem of computing PSNE, then moving to the more general problem of computing MSNE.

### 3.1   Finding a PSNE in $\alpha$-IDS games

In this subsection, we look at the complexity of determining a PSNE in $\alpha$-IDS games, and finding it if one exists. Our first result follows.

**Theorem 1** *Determining whether there is a PSNE in $n$-player SC+SS $\alpha$-IDS games is NP-complete.*

**Proof** *(Sketch)* We are going to reduce an instance of a 3-SAT variant into our problem. Each clause of the 3-SAT variant contains either only negated variables or only un-negated variables [15]. We

have an SC player for each clause and two SS players for each variable. The clause players invest if there exists a neighbor (its literal) that invests. For each variable $v_i$, we introduce two players $v_i$ and $\bar{v}_i$ with preference for mutually opposite actions. They invest if there exists a neighbor (its clause and $\bar{v}_i$) that does not invest. Figure 2 depicts the basic structure of the game. Nodes at the botton-row of the graph correspond to a variable, where the un-negated-variables-clauses and negated-variables-clauses are connected to their corresponding un-negated-variable and negated variable with bidirectional transfer probability $q$.

**Setting the parameters of the clause players.** Wlog, we can set the parameters to be identical for all clause players $i$: find $R_i > 0$ and $\alpha_i > 1 - p_i$ such that $(1-q)^2 > \Delta_i^{sc} > (1-q)^3$.

**Setting the parameters of the variables players.** Wlog, we can set the parameters to be identical for all variable players $i$: find $R_i > 0$ and $\alpha_i < 1 - p_i$ such that $1 > \Delta_i^{ss} > (1-q)$.

We now show that there exists a satisfiable assignment if and only if there exists a PSNE.

**Satisfiable assignment $\implies$ PSNE.** Suppose that we have a satisfiable assignment of the variant 3-SAT. This implies that every clause player is playing invest. Moreover, for each clause player, there must be some corresponding variable players that play invest. Given a satisfiable assignment, negated and un-negated variable players cannot play the same action. One of them must be playing invest and the other must be playing no-invest. The investing variable is best-responding because at least one of the players (namely its negation) is playing not invest. The not investing variable is best-responding because all of its neighbors are investing. Hence, all the players are best-responding to each other and thus we have a PSNE.

**PSNE $\implies$ satisfiable assignment.** (a) First we show that at every PSNE, all of the clause players must play invest. For the sake of contradiction, suppose that there is a PSNE in which there are some clause players that play no-invest. For the no-invest clause players, all of their variables must play no-invest at PSNE. However, by the best-response conditions of the variable players, if there exists a clause player that plays no-invest, then at least one of the variable players must play invest, which contradicts the fact that we have a PSNE. (b) We now show that at every PSNE, the un-negated variable player and the corresponding negated variable player must play different actions. Suppose that there is a PSNE, in which both of the players play the same action (i) no-invest or (ii) invest. In the case of no-invest (i), by their best-response conditions (given that at every PSNE all clause players play invest), none of the variables are best-responding so one of them must switch from playing no-invest to invest. In the case of invest (ii), again by the best-response condition, one of them must play no-invest. (c) Finally, we need to show that at every PSNE there must be a variable player that makes every clause player play invest. To see this, note that, by the clause's best-response condition, there must be at least one variable player playing invest. If there is a clause that plays invest when none of its variable players play invest, then the clause player would not be best-responding. □

### 3.1.1 SC $\alpha$-IDS games

What is the complexity of determining whether a PSNE exists in SC $\alpha$-IDS games (i.e. $\alpha_i > 1 - p_i$)? It turns out that SC players have the characteristics of following the actions of other agents. If there are enough SC players who invest, then some remaining SC player(s) will follow suit. This is evident from the safety function and the best-response condition. Consider the dynamics in which everybody starts off with no-invest. If there are some players that are not best-responding, then their best (dominant) strategy is to invest. We can safely change the actions of those players to invest. Then, for the remaining players, we continue to check to see if any of them is not best-responding. If not, we have a PSNE, otherwise, we change the strategy of the not best-responding players to invest. The process continues until we have reached a PSNE.

**Theorem 2** *There is an $O(n^2)$-time algorithm to compute a PSNE of any $n$-player SC $\alpha$-IDS game.*

Note that once a player plays invest, other players will either stay no-invest or move to invest. The no-investing players do not affect the strategy of the players that already have decided to invest. Players that have decided to invest will continue to invest because only more players will invest.

### 3.1.2 SS $\alpha$-IDS games

Unlike the SC case, an SS $\alpha$-IDS game may not have a PSNE when $n > 2$.

**Proposition 1** *Suppose we have an $n$-player SS $\alpha$-IDS game with $1 > \Delta_i^{ss} > (1 - q_{ji})$ where $j$ is the parent of $i$. (a) If the game graph is a directed tree, then the game has a PSNE. (b) If the game graph is a a directed cycle, then the game has a PSNE if and only if $n$ is even.*

**Proof** (a) The root of the tree will always play no-invest while the immediate children of the root will always play invest at a PSNE. Moreover, assigning the action invest or no-invest to any node that has an odd or even (undirected) distance to the root, respectively, completes the PSNE.

(b) For even $n$, an assignment in which any independent set of $\frac{n}{2}$ players play invest form a PSNE. For odd $n$, suppose there is a PSNE in which $I$ players invest and $N$ players do not invest, such that $I + N = n$. The investing players must have $I$ parents that do not invest and the non-investing players must have $N$ parents that play invest. Moreover, $I \leq N$ and $N \leq I$ implies that $I = N$. Hence, an odd $n$ cycle cannot have a PSNE. □

We leave the computational complexity of determining whether SS $\alpha$-IDS games have PSNE open.

### 3.2 Computing all NE in $\alpha$-IDS games

We now study whether we can compute *all* MSNE of $\alpha$-IDS games. We prove that we can compute all MSNE in polynomial time in the case of uniform-transfer SC $\alpha$-IDS games, and a subset of all MSNE in the case of SS and SC+SS games. A *uniform transfer $\alpha$-IDS game* is an $\alpha$-IDS game where the transfer probability to another players from a particular player is the same regardless of the destination. More formally, $q_{ij} = \delta_i$ for all players $i$ and $j$ ($i \neq j$). Hence, we have a complete graph with bidirectional transfer probabilities. We can express the overall safety function given joint mixed-strategy $\mathbf{x} \in [0,1]^n$ as $s(\mathbf{x}) = \prod_{i=1}^{n}[1-(1-x_i)\delta_i]$. Now, we can determine the best response of SC or SS player exactly based solely on the values of $\Delta_i^{sc}(1 - (1 - a_i)\delta_i)$, for SC, relative to $s(\mathbf{x})$; similarly for SS.

We assume, wlog, that for all players $i$, $R_i > 0$, $\delta_i > 0$, $p_i > 0$, and $\alpha_i > 0$. Given a joint mixed-strategy $\mathbf{x}$, we partition the players by type wrt $\mathbf{x}$: let $I \equiv I(\mathbf{x}) \equiv \{i \mid x_i = 1\}$, $N \equiv N(\mathbf{x}) \equiv \{i \mid x_i = 0\}$, and $P \equiv P(\mathbf{x}) \equiv \{i \mid 0 < x_i < 1\}$ be the set of players that, wrt $\mathbf{x}$, fully invest in protection, do not invest in protection, and partially invest in protection, respectively.

#### 3.2.1 Uniform-transfer SC $\alpha$-IDS games

The results of this section are non-trivial extensions of those of Kearns and Ortiz [8]. In particular, we can construct a polynomial-time algorithm to compute all MSNE of a uniform-transfer SC $\alpha$-IDS game, along the same lines of Kearns and Ortiz [8], by extending their Ordering Lemma (their Lemma 3) and Partial-Ordering Lemma (their Lemma 4). [1] Appendixes A.1 and B of the supplementary material contain our versions of the lemmas and detailed pseudocode for the algorithm, respectively. A running-time analysis similar to that for traditional uniform-transfer IDS games done by Kearns and Ortiz [8] yields our next algorithmic result.

**Theorem 3** *There exists an $O(n^4)$-time algorithm to compute all MSNE of an uniform-transfer $n$-player SC $\alpha$-IDS game.*

The significance of the theorem lies in its simplicity. That we can extend almost the same computational results, and structural implications on the solution space, to a considerably more general, and perhaps even more realistic, model, via what in hindsight were simple adaptations, is positive.

#### 3.2.2 Uniform-transfer SS $\alpha$-IDS games

Unlike the SC case, the ordering we get for the SS case does not yield an analogous lemma. Nevertheless, it turns out that we can still determine the mixed strategies of the partially-investing players in $P$ relative to a partition. The result is a Partial-Investment Lemma that is analogous to that of Kearns and Ortiz [8] for traditional IDS games. [2] For completeness, Appendix A.2 of the supplementary material formally states the lemma. We remind the reader that the significance and strength

of this non-trivial extension lies in its simplicity, and particularly when we note that the nature of the SS case is the complete opposite of the version of IDS games studied by Kearns and Ortiz [8].

Indeed, a naive way to compute all NE is to consider all of the possible combinations of players into the investment, partial investment, and not investment sets and apply the Partial-Investment Lemma alluded to in the previous paragraph to compute the mixed strategies. However, this would take $O(n^{ss}3^{n^{ss}})$ worst-case time to compute any equilibrium. So, how can we efficiently perform this computation? As mentioned earlier, SS players are less likely to invest when there is a large number of players investing and have "opposite" behavior as the SC players (i.e., the best response is flipped). Hence, imposing a "flip" ordering (Ordering 1) that is opposite of the SC case seems natural. If we assume such a specific ordering of the players at equilibrium, then we can compute all NE consistent with that specific ordering efficiently, as we discuss earlier for the SC case. Mirroring the SC $\alpha$-IDS game, we settle for computing all NE that satisfy the following ordering.

**Ordering 1** *For all $i \in I^{ss}, j \in P^{ss}$, and $k \in N^{ss}$,*

$$(1 - \delta_k)\Delta_k^{ss} \leq (1 - \delta_j)\Delta_j^{ss} < \Delta_j^{ss}$$
$$(1 - \delta_j)\Delta_j^{ss} \leq \Delta_j^{ss} \leq \Delta_i^{ss}$$
$$(1 - \delta_k)\Delta_k^{ss} \leq (1 - \delta_i)\Delta_i^{ss} \leq \Delta_i^{ss}$$

The first and last set of inequalities (ignoring the middle one) follow from the consistency constraint imposed by the overall safety function. The middle set of inequalities restrict and reduce the number of possible NE configurations we need to check. It is possible that the $(1-\delta_k)\Delta_k^{ss} > (1-\delta_j)\Delta_j^{ss}$ or $(1-\delta_k)\Delta_k^{ss} > (1-\delta_i)\Delta_i^{ss}$ at an NE, but we do not consider those types of NE. Our hardness results presented in the upcoming Section 3.2.4 suggest that, in general, computing all MSNE without any of the constraints above is likely hard. (See Algorithm 2 of the supplementary material.)

**Theorem 4** *There exists an $O(n^4)$-time algorithm to compute all MSNE consistent with Ordering 1 of an uniform-transfer $n$-player SS $\alpha$-IDS game.*

### 3.2.3 Uniform-transfer SC+SS $\alpha$-IDS games

For the uniform variant of the SC+SS $\alpha$-IDS games, we could partition the players into either SC or SS and modify the respective algorithms to compute all NE. Unfortunately, this is computationally infeasible because we can only compute all NE in polynomial time in the SC case. Again, if we settle for computing all NE consistent with Ordering 1, then we can devise an efficient algorithm. From now on, the fact that we are only considering NE consistent with Ordering 1 is implicit, unless noted otherwise. The idea is to partition the players into a class of SC and a class of SS players. From the characterizations stated earlier, it is clear that there are only a polynomial number of possible partitions we need to check for each class of players. Since the ordering results are based on the same overall safety function, the orderings of SC and SS players do not affect each other. Hence, wlog, starting with the algorithm described earlier as a based routine for SC players, we do the following. For each possible equilibrium configuration of the SC players, we first run the algorithm described in the previous section for SS players and then test whether the resulting joint mixed-strategy is a NE. This guarantees that we check every possible equilibrium combination. A running-time analysis yields our next result.

**Theorem 5** *There exists an $O(n_{sc}^4 n_{ss}^3 + n_{sc}^3 n_{ss}^4)$-time algorithm to compute all NE consistent with Ordering 1 of an uniform-transfer $n$-player SC+SS $\alpha$-IDS game, where $n = n^{sc} + n^{ss}$.*

### 3.2.4 Computing all MSNE of arbitrary $\alpha$-IDS games is intractable, in general

In this section, we prove that determining whether there exists a PSNE consistent with a partial-assignment of the actions to some players is NP-complete, even if the transfer probability takes only two values: $\delta_i \in \{0, q\}$ for $q \in (0, 1)$.

We consider the *pure-Nash-extension problem* [8] for binary-action $n$-player games that takes as input a description of the game and a *partial* assignment $\mathbf{a} \in \{0, 1, *\}^n$. We want to know whether there is a *complete* assignment $\mathbf{b} \in \{0, 1\}^n$ consistent with $\mathbf{a}$. Indeed, computing all NE is at least as difficult as the pure-Nash extension problem. Appendix C presents proofs of our next results.

Table 2: Level of Investment of SC+SS $\alpha$-IDS Games at Nash Equilibrium

| **High** $\frac{C_i}{L_i}$ | $\alpha_i \sim \mathcal{N}(0.4, 0.2)$ | | | $\alpha_i \sim \mathcal{N}(0.8, 0.2)$ | | | $\alpha_i \in [0, 1]$ | | |
|---|---|---|---|---|---|---|---|---|---|
| Datasets | %SS | %SC Invest | %SS Invest | %SS | %SC Invest | %SS Invest | %SS | %SC Invest | %SS Invest |
| Karate Club | 76.18 | 100.00 | 21.37 | 12.35 | 100.00 | 0.00 | 56.18 | 100.00 | 14.88 |
| Les Miserables | 75.45 | 100.00 | 17.93 | 11.82 | 99.85 | 0.67 | 55.06 | 99.40 | 14.84 |
| College Football | 75.65 | 100.00 | 15.47 | 11.57 | 100.00 | 0.00 | 55.39 | 100.00 | 13.46 |
| Power Grid | 75.47 | 97.76* | 19.38* | 12.82 | 98.79* | 2.13* | 55.01 | 97.31** | 15.90** |
| Wiki Vote | 75.55 | 97.46* | 17.87* | 12.78 | 98.92* | 2.06* | 55.02 | 97.00** | 14.75** |
| Email Enron | 75.29 | 95.97* | 19.91* | 12.53 | 97.92* | 2.24* | 54.78 | 94.39** | 16.84** |
| **Low** $\frac{C_i}{L_i}$ | $\alpha_i \sim \mathcal{N}(0.4, 0.2)$ | | | $\alpha_i \sim \mathcal{N}(0.8, 0.2)$ | | | $\alpha_i \in [0, 1]$ | | |
| Karate Club | 99.41 | 100.00 | 49.64 | 60.59 | 100.00 | 23.19 | 86.18 | 100.00 | 41.34 |
| Les Miserables | 98.96 | 100.00 | 51.17 | 59.22 | 100.00 | 28.34 | 85.71 | 100.00 | 49.26 |
| College Football | 98.87 | 100.00 | 60.42 | 61.48 | 100.00 | 28.30 | 86.35 | 100.00 | 54.87 |
| Power Grid | 98.68 | 99.13* | 49.45* | 59.41 | 98.81* | 28.66* | 85.20 | 99.13** | 45.07** |
| Wiki Vote | 98.62 | 98.30* | 46.50* | 59.89 | 97.38* | 27.54* | 85.01 | 98.51** | 44.45** |
| Email Enron | 98.73 | 97.96** | 49.80** | 59.85 | 96.48* | 29.32* | 84.94 | 98.0** | 44.72** |

*=0.001-NE, **=0.005-NE, %SS (%SC) = Percentage of SS (SC) players, $\mathcal{N}(\mu, \sigma^2)$ =normal distribution with mean $\mu$ and variance $\sigma^2$

**Theorem 6** *The pure-Nash extension problem for $n$-player SC $\alpha$-IDS games is NP-complete.*

A similar proof argument yields the following computational-complexity result.

**Theorem 7** *The pure-Nash extension problem for $n$-player SS $\alpha$-IDS games is NP-complete.*

Combining Theorems 6 and 7 yields the next corollary.

**Corollary 1** *The pure-Nash extension problem for $n$-player SC+SS $\alpha$-IDS games is NP-complete.*

## 4 Preliminary Experimental Results

To illustrate the impact of the $\alpha$ parameter on $\alpha$-IDS games, we perform experiments on randomly-generated instances of $\alpha$-IDS games in which we compute a possibly approximate NE. Given $\epsilon > 0$, in an approximate $\epsilon$-NE each individual's unilateral deviation cannot reduce the individual's expected cost by more than $\epsilon$. The underlying structures of the instances use network graphs from publicly-available, real-world datasets [6, 16–20]. Appendix D of the supplementary material provides more specific information on the size of the different graphs in the real-world dataset. The number of nodes/players ranges from $34$ to $\approx 37K$ while the number of edges ranges from $78$ to $\approx 368K$. The table lists the graphs in increasing size (from top to bottom). To generate each instance we generate (1) $C_i/L_i$ where $C_i = 10^3 * (1 + \text{random}(0, 1))$ and $L_i = 10^4$ (or $L_i = 10^4/3$) to obtain a low (high) cost-to-loss ratio and $\alpha_i$ values as specified in the experiments; (2) $p_i$ such that $\Delta_i^{sc}$ or $\Delta_i^{ss}$ is $[0, 1]$; and (3) $q_{ji}$'s consistent with probabilistic constraints relative to the other parameters (i.e. $p_i + \sum_{j \in Pa(i)} q_{ji} \le 1$). On each instance, we initialize the players' mixed strategies uniformly at random and run a simple gradient-dynamics heuristic based on regret minimization [21–23] until we reach an ($\epsilon$) NE. In short, we update the strategies of all non-$\epsilon$-best-responding players $i$ at each round $t$ according to $x_i^{(t+1)} \leftarrow x_i^{(t)} - 10 \times (M_i(1, \mathbf{x}_{Pa(i)}^{(t)}) - M_i(0, \mathbf{x}_{Pa(i)}^{(t)}))$. Note that for $\epsilon$-NE to be well-defined, all $M_i$s' values are normalized. Given that our main interest is to study the structural properties of arbitrary $\alpha$-IDS games, our hardness results of computing NE in such games justify the use of a heuristic as we do here. (Kearns and Ortiz [8] and Chan et al. [7] also used a similar heuristic in their experiments.). Table 2 shows the average level of investment at NE over ten runs on each graph instance. We observe that higher $\alpha$ values generate more SC players, consistent with the nature of the game instances. Almost all of the SC players invest while most of the SS players do not invest, regardless of the number of players in the games and the $\alpha$ values. This makes sense because of the nature of the SC and SS players. Going from high to low cost-to-loss ratio, we see that the number of SS players and the percentage of SS players investing at a NE increase across all $\alpha$ values. In both high and low cost-to-loss ratio cases, we see a similar behavior in which the majority of the SS players do not invest ($\approx 50\%$).

**Acknowledgments**
This material is based upon work supported by an NSF Graduate Research Fellowship (first author) and an NSF CAREER Award IIS-1054541 (second author).

## Footnotes

[1]Take their $R_i/p_i$'s and replace them with our corresponding $\Delta_i^{sc}$'s.

[2]Take their Lemma 4 and replace $R_i/p_i$ there by $\Delta_i^{ss}$ here, and replace the expression for $V$ there by $V \equiv [\max_{k \in N}(1 - \delta_k)\Delta_k^{ss}, \min_{i \in I}\Delta_i^{ss}]$.

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
