[Supplementary Material]

# Computing Nash Equilibria in Generalized Interdependent Security Games: Supplementary Material

**Hau Chan**      **Luis E. Ortiz**
Department of Computer Science, Stony Brook University
{hauchan,leortiz}@cs.stonybrook.edu

## A   Uniform-transfer $\alpha$-IDS games: lemmas

Under the uniform-transfer $\alpha$-IDS games, the overall safety function, given a joint mixed-strategy $\mathbf{x} \in [0,1]^n$, is $s(\mathbf{x}) = \prod_{i=1}^{n}[1 - (1 - x_i)\delta_i]$. Now, we can determine the best response of a SC (or SS) player exactly based solely on the value of $\Delta_i^{sc}(1 - (1 - a_i)\delta_i)$ (or $\Delta_i^{ss}(1 - (1 - a_i)\delta_i)$) relative to $s(\mathbf{x})$.

In the following, we assume, without loss of generality, that for all players $i$, $R_i > 0$, $\delta_i > 0$, $p_i > 0$, and $\alpha_i > 0$. Given a joint mixed-strategy $\mathbf{x}$, we partition the players by type w.r.t. $\mathbf{x}$: let $I \equiv I(\mathbf{x}) \equiv \{i \mid x_i = 1\}$, $N \equiv N(\mathbf{x}) \equiv \{i \mid x_i = 0\}$, and $P \equiv P(\mathbf{x}) \equiv \{i \mid 0 < x_i < 1\}$ be the set of players that fully invest in protection, do not invest in protection, and partially invest in protection, respectively.

### A.1   Uniform-transfer SC $\alpha$-IDS games

**Lemma 1 (Ordering Lemma)** *Suppose $\mathbf{x}$ is a NE of a uniform-transfer SC $\alpha$-IDS game. Then for any $i \in I$ (investing players), any $j \in P$ (partially investing players), and any $k \in N$ (not investing players), then*

$$\Delta_i^{sc} \le \Delta_j^{sc}$$
$$\Delta_i^{sc} \le (1 - \delta_k)\Delta_k^{sc} < \Delta_k^{sc}$$
$$(1 - \delta_j)\Delta_j^{sc} \le (1 - \delta_k)\Delta_k^{sc}$$

**Proof**  The inequalities follow immediately by using the overall safety function to compare the players in $I$, $P$, and $N$. □

The following Lemma specifies the strategies of the players in the partially investing set.

**Lemma 2 (Partial Investment Lemma)** *Suppose $\mathbf{x}$ is a NE of a uniform-transfer SC $\alpha$-IDS game. For any $j \in P$,*

1. *If $|P| = 1$, then $x_j \in \frac{1}{\delta}\left(\frac{1}{\Delta_j^{sc}}V - (1 - \delta_j)\right)$*

2. *if $|P| > 1$, then $x_j = \frac{1}{\delta}\left(\frac{1}{\Delta_j^{sc}}V^* - (1 - \delta_j)\right)$*

*where $V = [\max_{i \in I} \Delta_i^{sc}, \min_{k \in N}(1 - \delta_k)\Delta_k^{sc}]$ and $V^* = \left(\frac{\prod_{j \in P} \Delta_j^{sc}}{\prod_{k \in N}(1 - \delta_k)}\right)^{\frac{1}{|P|-1}}$.*

**Proof**  Suppose that $|P| = 1$. By the best-response condition $\Delta_j^{sc} = \prod_{l \in N}(1 - \delta_l)$. Moreover

$$\forall\, i \in I, \Delta_i^{sc} \le (1 - (1 - x_j)\delta_j) \prod_{l \in N}(1 - \delta_l)$$

and

$$\forall\, k \in N, (1 - \delta_k)\Delta_k^{sc} \geq (1 - (1 - x_j)\delta_j) \prod_{l \in N}(1 - \delta_l).$$

If we solve for $x_j$, we can obtain the values that $x_j$ can take at an equilibrium.

Suppose that $|P| > 1$. By the best-response condition

$$\Delta_j^{sc} = \prod_{p \in P - \{j\}}(1 - (1 - x_p)\delta_p) \prod_{l \in N}(1 - \delta_l)\, \forall j \in P.$$

Furthermore, for $j \in P$,

$$\prod_{k \in P - j}\Delta_k^{sc} = (1 - (1 - x_j)\delta_j)^{|P|-1}(\prod_{p \in P - j}(1 - (1 - x_p)\delta_p))^{|P|-2}(\prod_{l \in N}(1 - \delta_l))^{|P|-1}$$

It follows that

$$\frac{\prod_{k \in P - j}\Delta_k^{sc}}{(\prod_{p \in P - j}(1 - (1 - x_p)\delta_p))^{|P|-2}(\prod_{l \in N}(1 - \delta_l))^{|P|-1}} = (1 - (1 - x_j)\delta_j)^{|P|-1}$$

$$\frac{\prod_{k \in P}\Delta_k^{sc}}{(\prod_{p \in P - j}(1 - (1 - x_p)\delta_p))^{|P|-1}(\prod_{l \in N}(1 - \delta_l))^{|P|}} = (1 - (1 - x_j)\delta_j)^{|P|-1}$$

$$(\frac{\prod_{k \in P}\Delta_k^{sc}}{(\prod_{p \in P - j}(1 - (1 - x_p)\delta_p))^{|P|-1}(\prod_{l \in N}(1 - \delta_l))^{|P|}})^{\frac{1}{|P|-1}} = (1 - (1 - x_j)\delta_j)$$

$$(\frac{\prod_{k \in P}\Delta_k^{sc}}{\prod_{l \in N}(1 - \delta_l)})^{\frac{1}{|P|-1}}\frac{1}{(\prod_{p \in P - j}(1 - (1 - x_p)\delta_p))\prod_{l \in N}(1 - \delta_l)} = (1 - (1 - x_j)\delta_j)$$

$$(\frac{\prod_{k \in P}\Delta_k^{sc}}{\prod_{l \in N}(1 - \delta_l)})^{\frac{1}{|P|-1}}\frac{1}{\Delta_j^{sc}} = (1 - (1 - x_j)\delta_j)$$

The result follows from solving for $x_j$. $\qquad\square$

## A.2 Uniform-transfer SS $\alpha$-IDS games

**Lemma 3** *(Partial Investment Lemma) Suppose $\boldsymbol{x}$ is a NE of a uniform-transfer SS $\alpha$-IDS game. For any $j \in P$,*

1. *If $|P| = 1$, then $x_j \in \frac{1}{\delta}(\frac{1}{\Delta_j^{ss}}V - (1 - \delta_j))$*

2. *if $|P| > 1$, then use Lemma 2 part 2.*

*where $V = [\max_{k \in N}(1 - \delta_k)\Delta_k^{ss}, \min_{i \in I}\Delta_i^{ss}]$.*

**Proof** The proof is similar to the one in Lemma 2. $\qquad\square$

## B   Pseudocode for computing all NE in uniform-transfer $\alpha$-IDS games

This section contains the pseudocode of the algorithms described in the main body of the paper. In particular, Algorithm 1 and Algorithm 2 are algorithms to compute all NE in uniform-transfer SC $\alpha$-IDS games and uniform-transfer SS $\alpha$-IDS games, respectively. The subroutine **TestNash** of Algorithm 1 is outlined in Algorithm 3. The subroutine **TestNash** of Algorithm 2 can be constructed similarly from Algorithm 3 where it will use Lemma 3.

The running time of Algorithm 1 and Algorithm 2 is $O(n_{sc}^3)$ and $O(n_{ss}^3)$, respectively, where the **TestNash** subroutine takes $O(n)$, and line 7 of the algorithms runs in $O(n(1 + 2 + ... + n) = O(n^3)$ times for $n = n_{sc}$ or $n = n_{ss}$.

---
**Algorithm 1:** Compute all Nash equilibria of SC $\alpha$-IDS games
---
**Input** : An instance of $n$-players SC $\alpha$-IDS Game
**Output**: S - The set of all Nash equilibria of the input game
1  $I \leftarrow \{1, ..., n\}, P \leftarrow \{\}, N \leftarrow \{\}$
2  $S \leftarrow \textbf{TestNash}(I, P, N)$
3  Order $(i_1, i_2, ..., i_n)$ such that $\Delta^{sc}_{i_1} \geq ... \geq \Delta^{sc}_{i_n}$
4  **foreach** $k = 1, ..., n$ **do**
5      $P \leftarrow P \cup \{i_k\}, I \leftarrow I - \{i_k\}, N \leftarrow \{\}, S \leftarrow S \bigcup \textbf{TestNash}(I, P, N)$
6      Let $P' \leftarrow P$ and order $(j_1, ..., j_k)$ such that $(1 - \delta_{j_1})\Delta^{sc}_{j_1} \geq ... \geq (1 - \delta_{j_k})\Delta^{sc}_{j_k}$
7      **foreach** $m = 1, ..., k$ **do**
8          $N \leftarrow N \cup \{j_m\}, P' \leftarrow P' - \{j_m\} \ S \leftarrow S \bigcup \textbf{TestNash}(I, P', N)$
9      **end foreach**
10 **end foreach**
11 **return** $S$
---

---
**Algorithm 2:** Compute All Nash Equilibrium of SS consistent with Ordering 1
---
**Input** : An instance of $n$-players SS $\alpha$-IDS Game
**Output**: S - A set of all Nash Equilibrium that is consistent with Ordering 1
1  $I \leftarrow \{\}, P \leftarrow \{\}, N \leftarrow \{1, ..., n\}$
2  $S \leftarrow \textbf{TestNash}(I, P, N, S)$
3  Order $(i_1, i_2, ..., i_n)$ such that $(1 - \delta_{i_1})\Delta^{ss}_{i_1} \geq ... \geq (1 - \delta_{i_n})\Delta^{ss}_{i_n}$
4  **foreach** $k = 1, ..., n$ **do**
5      $P \leftarrow P \cup \{i_k\}, N \leftarrow N - \{i_k\}, I \leftarrow \{\}, S \leftarrow \textbf{TestNash}(I, P, N, S)$
6      Let $P' \leftarrow P$ and order $(j_1, ..., j_k)$ such that $\Delta^{ss}_{j_1} \geq ... \geq \Delta^{ss}_{j_k}$
7      **foreach** $m = 1, ..., k$ **do**
8          $I \leftarrow I \cup \{j_m\}, P' \leftarrow P' - \{j_m\} \ S \leftarrow \textbf{TestNash}(I, P', N, S)$
9      **end foreach**
10 **end foreach**
11 **return** $S$
---

---
**Algorithm 3:** **TestNash** subroutine
---
**Input** : A partition of the players into I, P, and N
**Output**: S - The set of all Nash equilibria consistent with the input partition
1  $\forall i \in I, x_i \leftarrow 0, \forall k \in N, x_k \leftarrow 0$
2  **if** $|P| = 1$ *and* $j \in P$ *(Lemma 2 Part 1)* **then**
3      Let $U' = U \cap (0, 1)$
4      **if** $\Delta^{sc}_j = \prod_{k \in N}(1 - \delta_k)$ *and* $U' \neq \emptyset$ **then**
5          $S \leftarrow \{\mathbf{y} \mid y_j \in U', \mathbf{y}_{-j} = \mathbf{x}_{-j}\}$
6      **end if**
7  **else** Lemma 2 Part 2
8      $\forall j \in P$, compute $x_j$
9      **if** $\mathbf{x}$ *is an MSNE of the input game* **then**
10         $S \leftarrow \{\mathbf{x}\}$
11     **end if**
12 **end if**
13 **return** $S$
---

## C Computing all MSNE of arbitrary $\alpha$-IDS games

### C.1 Proof sketch of Theorem 6

In the following, we will show that determining whether there exists a PSNE consistent with a partial-assignment of the actions to some players is NP-complete, even if the transfer probability takes only two values: $\delta_i \in \{0, q\}$ for $q \in (0, 1)$.

Figure 1: 3-SAT-induced $\alpha$-IDS game graph

More specifically, this section contains proof to show that the pure-Nash extension problem for $n$-player SC $\alpha$-IDS and $n$-player SS $\alpha$-IDS games are NP-complete.

The *pure-Nash-extension problem* [1] for binary-action $n$-player games that takes as input a description of the game and a *partial* assignment $\mathbf{a} \in \{0, 1, *\}^n$. We want to know whether there is a *complete* assignment $\mathbf{b} \in \{0, 1\}^n$ consistent with $\mathbf{a}$.

**Theorem 1** *(Theorem 6 of the Main Paper) The pure-Nash extension problem for $n$-player SC $\alpha$-IDS games is NP-complete.*

**Proof** (Sketch) We reduce from Monotone 1 in 3-SAT [2]. The big idea is to consider a bipartite graph structure (Figure C.2) between the clauses and the variables (all direct edges from variables to the corresponding clause players with transfer probability $q > 0$). We introduce two players ($a_i$ and $b_i$) for each clause $i$. Player $a_i$ invests if at least one of its variable players invest. Player $b_i$ invests if at least two of its variable players invest. For clause players $a_i$ and $b_i$, we find $R_j > 0$ and $\alpha_j > 1 - p_j$ for $j \in \{a_i, b_i\}$ such that $(1-q)^2 > \Delta_{a_i}^{sc} > (1-q)^3$ and $(1-q) > \Delta_{b_i}^{sc} > (1-q)^2$, respectively. The variable players would be indifferent between invest and not invest. For each variable player $i$, we just need to make sure that $\Delta_i^{sc} = 1$ (or $R_i = p_i$). Finally, we give partial assignments to the clauses player where $a_i$ invests and $b_i$ not invest to guarantee that exactly one invests and solution to Monotone 1 in 3-SAT. $\square$

### C.2 Proof sketch of Theorem 7

**Theorem 2** *(Theorem 7 of the Main Paper) The pure-Nash extension problem for $n$-player SS $\alpha$-IDS games is NP-complete.*

**Proof** (Sketch) This is similar to the proof of Theorem 6 except the best-response of the players and using the game graph as in Figure C.2. For each clause $i$, we introduce two clauses players $a_i$ and $b_i$. Player $a_i$ invests if at least two of its variable players do not invest. Player $b_i$ invests if at least three (or all three) of its variable players do not invest. Mainly, find $R_j > 0$ and $\alpha_j < 1 - p_j$ for $j \in \{a_i, b_i\}$ such that $(1-q) > \Delta_{a_i}^{ss} > (1-q)^2$ and $(1-q)^2 > \Delta_{b_i}^{ss} > (1-q)^3$. The variable players would be indifferent between invest and not invest. We give partial assignment to the clauses player where $a_i$ invests and $b_i$ not invest to guarantee that exactly one invests. $\square$

## D  On real-world graph dataset used for the experiments

Table 1 shows the exact number of nodes and edges for each of the graphs from the real-world datasets we used for our experiments.

Table 1: Exact number of nodes and edges for different real-world graphs

| Graph | Nodes | Edges |
|---|---|---|
| Karate Club | 34 | 78 |
| Les Miserables | 77 | 254 |
| College Football | 115 | 613 |
| Power Grid | 4941 | 6594 |
| Wiki Vote | 7115 | 103689 |
| Email Enron | 36692 | 367662 |