[Reviews · NeurIPS 2014]

Submitted by Assigned_Reviewer_10

The paper presents novel results in computing, and equilibrium characters of interdependent security games, for different ranges of the transfer probabilities (alpha-IDS).

The paper presents the problem very clearly, and describes the novelty of the results. To the best of my understanding, the paper is technically correct. The main criticism is that it is not clear how the equilibrium is computed for the experiments - more explanation, and more analyses, is required.
Summary: The paper presents novel and interesting results in computing, and equilibrium characters of interdependent security games, for different ranges of the transfer probabilities (alpha-IDS), and different types of players (Strategic substitutes and strategic complementaries).

Submitted by Assigned_Reviewer_23

The paper introduces a new type of game, generalized interdependent security games, based on Kearns and Ortiz 2004. Several new complexity results are derived and proven for this class of games and some preliminary experimental investigation is given.

This is a dense technical paper with many new (but quite specific) contributions for this new class of games. The problem is well-motivated, and the paper reads smoothly in sections 1 and 2. Section 3 (Computational results of alpha-IDS games) is difficult to follow. It feels like there is a lot, possibly too much, that the authors tried to fit into a single paper here and the main contribution/story gets a bit lost amidst the lemmas and proofs. The results are new and should be of interest to (at least) the same audience as that of [8], but the scope of the content does still feel a bit narrow. The experiments section is one long paragraph that lacks structure and polish. There is no conclusion nor much discussion of these results.

Questions: Is \Delta_i^sc = \Delta_i^ss? Why two different terms, for clarity of the context? In proof of Theorem one, "its clause" and "its literal": why are these singular? Fig2 shows variables can belong to several clauses. Also, I cannot explain the bounds (1-q)^2 > \Delta_i^sc > (1-q)^3 and 1 > \Delta_i^ss > (1-q); these seems inconsistent with the text and best response definitions, is there an error here?

Minor points:
- your's -> yours
- references, several missed capitalization (nash -> Nash)

** Added after response. I appreciate the clarifications made by the authors and pointers to previous research. These games seem to have been more widely-studied than I was first led to believe after my first read of the paper. I encourage the authors to make this more clear if possible. I understand that space is a limiting factor, but the second half of the paper could certainly use some "smoothing", and especially some more discussion on these results could help the impact this work will have, if accepted.

Summary: The paper introduces generalized interdependent security games and many new significant complexity results for finding different types of equilibria within these games. The scope of the paper is somewhat narrow and presentation suffers somewhat from the density of its content, which could negatively affect its impact.

Submitted by Assigned_Reviewer_42

In this work, authors study the computational complexity of computing Nash Equilibria in a generalized version of Interdependent security (IDS) games. In their generalized model, the investment in security can partially reduce the indirect risk of becoming exposed through neighbors. In other words, if an agent decides to fully invest in security, this action not only reduces his direct risk, but also reduces his transfer risk from his neighbors. A parameter \alpha_i captures the extent to which the transfer risk of agent i reduces with his decision to invest. Depending on the \alpha values, players can exhibit strategic complementarity (SC) or substitutability (SS). Authors consider 3 different cases: one in which all players are SC, one in which all players are SS, and one in which players are a combination of the two (SC+SS) types. For each case authors look at the problem of computing pure NE and mixed NE. For the case when all the players are SC, authors present a polynomial time algorithm to compute a pure NE. For the computation of mixed NEs, authors have to make additional assumption (i.e. uniform transfer probabilities and some ordering constraints) to be able to present polynomial algorithms. The rest of the results presented in the paper are mainly negative results.

I find this work an interesting extension of the previous work on interdependent security games. The exposition of the paper is clear. However while the results are novel, most of them (as one would expect) are negative.
Summary: I think it would be nice if authors could motivate their problem better by discussing (in more detail) situations where their model is applicable. Also I would like to hear authors comments about the necessity of computing *all* NEs in the IDS games.
Author Feedback
Author rebuttal: We would like to thank the reviewers for taking the time and effort to review our paper, including pointing out several typos.

We welcome any constructive criticism about the clarity of the presentation. Please bear in mind the space constraints when providing general suggestions.
Therefore, to accommodate any suggested/requested content, we would need to remove or condense another content. For example, we fully sympathize with the reviewers’ suggestions to expand the paper along a variety of topics. Any specific suggestions about how to introduce additional examples, explanations, or discussions, while maintaining clarity in the exposition, not making the presentation any denser, and respecting the space constraints, would be extremely useful. The same holds for comments about the apparently high density of the presentation.

We emphasized the theoretical results in the presentation because our main objective in the experimental section was to convey two main messages. First, that we can compute (epsilon) NEs of alpha-IDS games efficiently using a simple gradient-dynamics heuristic based on regret minimization. Second, that in the NEs that we computed, we observed that the ss players have the tendency of not investing and the sc players have the opposite behavior, which is consistent with what one would expect from sc and ss players. We thought that our brief experiment section achieved our objectives.

We would be grateful for specific feedback on how to strike the right balance between the presentation of the theoretical results (Section 3) and experimental results (Section 4), while keeping clarity, quality, and space constraints.

Regarding reviewer 23’s first question, yes, the mathematical expression for \Delta_i^{sc} and \Delta_i^{ss} is the same, so they are syntactically equivalent. They have different semantics, and we wanted to keep those separate interpretations, in terms of players’ behavior, clear. The sc and ss superscripts indicate the types of the players: strategic complementary (sc) and strategic substitutability (ss). The sc and ss players have different best-response conditions where their strict inequalities of the best-response conditions are “reverse”. (Please see lines 184-191 for the definition of the best-response correspondence.)

Regarding the proof of Theorem 1, we agree that using “a” instead of “its” would have been clearer; we apologize for any misunderstanding.

Having explained the meaning of \Delta_i^{sc} and \Delta_i^{ss} above, let us address the reviewer’s last question and explain the meaning of the bounds.

The bound for an sc player, (i.e., a clause), given by the condition (1-q)^2 > \Delta_i^{sc} > (1-q)^3, says that if at least one of its neighboring players (i.e., any of its literals) plays invest, then the sc player will play the action invest (in other words, the clause is satisfied). Otherwise, the sc player will play the action not-invest.

The bound for an ss player (i.e., a literal), given by the condition 1 > \Delta_i^{ss} > (1-q), says that if all of its neighbors (i.e., all clauses in which the literal appears, along with its negated literal) plays invest, then the ss player would not invest. Otherwise, the ss player will play the action invest.

We are unclear as to what a “narrow scope” means in the context of our paper. Certainly, we are working with the generalized version of a specific class of games. But the scope of those games, in terms of applicability and audience, is not narrow, as evidenced by the variety of papers on the subject. We included some references in our paper (see, e.g., Heal and Kunreuther, 2003, 2002, 2004, 2005; Kearns and Ortiz, NIPS 2004; Gkonis and Psaraftis, 2010). One can view uniform transfer alpha-IDS games, which generalized a similar class introduced by Kearns and Ortiz [NIPS 2004], as a special type of summarization game [Kearns and Mansour, 2002]. Our general algorithmic/computational approach may also extend to those games. There is even a recent survey devoted to IDS games [Laszka, Felegyhazi, and Buttyann, 2012]. Hence, the audience with potential interest in our work goes beyond that of graphical games [Kearns, Littman, and Singh, 2001].

Regarding reviewer 42’s question on “the necessity of computing all NEs” in alpha-IDS games, we want to point out that our goal is to understand the (rational) behavior of the players and all possible globally stable outcomes that such behavior may induce on the system. In the context of a non-cooperative security setting, computing all NEs is essential: missing a possible stable outcome may have dire consequences. We realize that the typical work in computational game theory mostly focuses on the computation of a single NE, which may be sufficient in some cases, but is not very meaningful in ours. Uncertainty about which outcome may occur forces the need to compute all NEs. We want to know and reason about anything that may happen, so that we can take it into consideration on any analysis, or even potential interventions.

As a side note, we originally had a more specific example but decided to use an adapted version of a simpler, more standard IDS example. The example we considered using was that of flu vaccination. In this example, agents decide whether or not to take the flu vaccine as protection against the flu given that the flu can be transferred from one agent to another. Of course, flu vaccine is not 100% effective; in fact, based on a report by Centers for Disease Control and Prevention (CDC), the effectiveness of flu vaccine for the 2013-2014 season was 60%). In this case, the effectiveness of the flu vaccine becomes the alpha parameter.

We hope our answers above will help the reviewers in the final evaluation of our submission.